# Cosmology, Faith, Architecture—A Temple under the Sky: The Church of Saint Maximilian Kolbe in Varese

Luca Placci 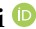

Department of Architecture, University of Florence, 50121 Firenze, FI, Italy; luca.placci@unifi.it

**Abstract:** In my article, I propose a reflection about the Catholic church of Saint Maximilian Kolbe in Varese, Italy, designed by the architect Justus Dahinden at the end of the millennium. Despite the fact that this original sacred space has been imagined by a well-known designer, it still remains a neglected case study. In detail, the present research is about the method by which the architect included the divine element into contemporary architecture and how he facilitated the encounter with the transcendent. The first step focused on the assessment of unpublished materials, such as the architect's early plan drafts, the executive drawings and the correspondence between the client and the designer. The following study was on Dahinden's scripts and publications. In the second stage, I analyzed the space under the lens of the hermeneutical approach to highlight the importance of the proven experiences in the building, which is distinguished for its holistic qualities. Furthermore, symbolism plays a relevant role in communicating the evangelical message here, and it seems that Dahinden brought it to the extreme consequence; the entire building, the sequence of its spaces and its details strongly evoke a universal dimension, which pretends to go beyond the dogmatism which marks the traditional religious architecture.

**Keywords:** sacred space; contemporary architecture; Justus Dahinden; transcendent; Varese; Italy





## 1. Introduction

A monumental cupola arises out of the blue when we walk along the avenue that connects the Varese city center to Sacro Monte (UNESCO site since 2003). A ring of water surrounding its base makes it dramatically fascinating, which would not have been anomalous if the structure had been stood on a drum, as it usually is. Through its stereometric shape, it strengthens its presence in the urban landscape. Even the pure white tone of the chromium-plated steel external surface—letting it resemble a shining "armored shell"—evokes an oversized igloo or a prehistoric remnant of the glacier age. The amorphous growth of the suburbs of the medium-sized town of Varese, which is 30 miles north of Milan, is banned by the unequivocal presence of the building, which does not remain unnoticed. It is as if the dome strongly declares that it distances itself from the low-value environment in which it is located. Nevertheless, it does not impose itself as a threatening profile. That is the reason why we get closer. We are in front of the Catholic parish church dedicated to the Polish Franciscan monk Saint Maximilian Kolbe (1894–1941), who perished in Auschwitz under Nazi occupation in 1941.

This specific architectonic episode gives us the chance to enlarge the perspective on the ecclesiastical design which occurred at the end of the last century. The objective of the present article is to analyze the architectonic approach followed by its designer and investigate the way he included the spiritual qualities into this sacred building to facilitate an encounter with God.

The words of Raoul Le Chauve resonate as familiar to us when, in the 11th century, the French friar affirmed that "one would have believed that the world, throwing away its ancient vestments, was adorned with a white mantle of new churches" (Le Chauve, *Historiae*, liber III, 3). This description utterly fits with the architectural context of the last

five decades if we consider that in Europe alone, hundreds of new white-plastered churches have been built. The candid Lutheran temples designed by Juha Leiviskä in Finland, the parish church of Dio Misericordioso in Rome drawn by Richard Meier, the church of Santa Maria de Canaveses erected by Alvaro Siza in Portugal and the only Catholic one created by Alvar Aalto in the Italian village of Riola testify to the new tendencies.

The church of Saint Maximilian Kolbe is not an exception in this scenario. It stands in line with the majority of the other interventions that prove the evolution of contemporary ecclesiastical architecture, although in the meantime, it shows great expressiveness, and it differs from the other sacred buildings in the experiential qualities of its space.

Unquestionably, each of these types of architecture takes into account the directives established during the Second Vatican Council (1962–1965), which was a turning point in the Catholic world. In addition, for the theological themes dealt with, the Council gave new impetus to reforms concerning religious buildings, as these had been extensively analyzed in the recent years (Benedetti 2000). Among the designers engaged in the unanimous effort to create new spaces for worship, the architect Justus Dahinden gave a fundamental contribution to the international scene (Dahinden 1987, 2005)[1]. His long experience in designing more than 20 ecclesiastical buildings in Europe, Africa and Asia convinced the client (a commission composed of 26 members) to give him the assignment for the new temple, as reported in the letter sent to him by the local priest don Giovanni Brigatti. The latter wished he could contribute to realizing a building which would have been a "segno di bellezza moderna" (translation: "an emblem of modern beauty")[2].

The church in Varese acts as an interesting case of late millennium design that took place in Italy between 1987 and 1993. The church was consecrated by Monsignor Carlo Maria Martini on 27 October 1996, although it had been in use since December 1993, when the very first service was celebrated.

Dahinden's involvement in the plan is reported below in order to illustrate his view on sacred architecture at the end of the last century (Brigatti 1997b).

## 2. The Commission and the Social Context

Initially, it is necessary to introduce another character who had a role in the narration: bishop Giuseppe Arosio of the Metropolitan Archdiocese of Milan (Brigatti 1997b, p. 107). As a young priest, he demonstrated an unconventional interest in modern sacred architecture, so in the 1950s, he spent some weeks traveling around Central Europe to detect the latest trends in modern architecture. That aside, he was one of the first Italians to set foot in Notre-Dame du Haut when Le Corbusier's masterpiece had just been opened. During his tour, he not only had the chance to draw upon prime architectural material, but he also met the designers who were involved in the reconstruction of the new churches after the damage caused by the war. In particular, the encounter with Dahinden was revealed to be of great poignancy. They were both born in 1925, and soon they became friends (Arosio 2000). Thereupon, when in the 1970s the plan of a new church in Monza near Milan—where Arosio was the pastor of the parish—was required, he immediately recommended hiring the Swiss architect for the task as well (Caligaris 1977). At that time, it was quite rare to invite a foreign designer to plan a Catholic church, and at first, the architect from the Alps was welcomed skeptically, which happened to Alvar Aalto in Riola in the same years (Gresleri and Gresleri 2004). Nevertheless, the suspicion vanished as soon as Dahinden gave birth to one of the most original architectonic episodes in the Milanese archdiocese (De Stefano 1983, p. 518).

This occurred in Varese, when the demand for a bigger place destined for worship compelled don Giovanni Brigatti to ask to the Ufficio delle Chiese Nuove (Office for the New Churches) to intervene, but not before having the community involved. Truthfully, in spring of 1985, the pastor invited 3500 members of the parish to fill out a questionnaire, in which he asked how they would have felt involved in the plan for the new church and parish center, whether they had any advice and, more relevant, how they imagined the new temple. The options were the following:

"(a) solo utile al servizio religioso ma un po' usuale

(b) anche suggestiva, in armonia con il territorio, seppur più costosa?"[3]

Unfortunately, not all the questionnaires are collected in the parish archive, so we can only presume that through this egalitarian method, the majority of the parishioners supported the mission and affirmed of being favorable to building a modern and beautiful church, as truly happened. The previous building was no longer able to accommodate the congregation, so Monsignor Arosio, who at that time was the director of the diocesan bureau, did not hesitate to solicit Justus Dahinden for help[4]. The architect accepted the assignment (October 1987) and in June 1988 he was ready to exhibit the draft during a meeting in Varese, which numerous citizens took part in, as expectations regarding his task were high (Brigatti 1997b). In February 1989, the project was forwarded to the municipality office, and the ceremony for the laying of the first stone took place on 17 June 1990. In addition to the church itself, the masterplan also included the chapel for the weekly services with the baptismal font, the vestry, the parish and meeting spaces in accordance with the instructions mentioned in the documents that had been drawn up during the Second Vatican Council (Figure 1). In fact, according to it, the neighborhood churches were supposed either to offer spaces for worship and spiritual recollection or provide areas for socializing, which were missing in the district (Figure 2).

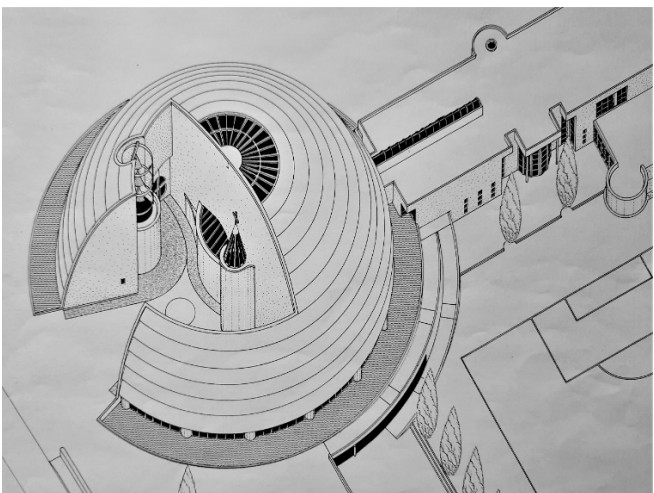

**Figure 1.** Axonometric plan of the church with the adjoining volume of the rectory on the north side.

A permanent parish replaced previous inappropriate venues, with the aim of sorting out the absence of places for non-religious activities but also with the goal of realizing poles of identification for the proximity. As a matter of fact, architect Dahinden was inclined to deem that "abstract ideologies [ . . . ] are never the keys to solving architectural problems" (Dahinden 2005, p. 20). This belief was at the basis of his design philosophy, which aimed to produce the gestalt by creating spaces in which the devotee felt at ease without resorting to preconceived models, but rather by finding solutions that varied from time to time, depending on the context.

However, there are some guidelines that must be respected in contemporary architecture:

1. An object serving a purpose;
2. A perceived medium;
3. A spiritual manifestation (Dahinden 2005, p. 77).

All of the above were implemented, in particular in church projects, in which the last point prevails. The spiritual strength had to be perceived through matter and form, so the architect had to face a challenge that he willingly accepted each time.

Moreover, thanks to his extensive practice in planning, he was aware of the anthropological quality of architecture as a holistic service to the believers. Humanity and spirit were

highly taken into account by him. Modern churches had to express an exhortative message rather than be awe-inspiring; the old symmetrical rigor and the common overabundance of decorations would repel devotees instead of attracting them (Dahinden 1987, p. 43). Dogmatic architecture no longer served for prayer, but it had to be a welcoming shelter, a place of stasis from the daily frenzy.

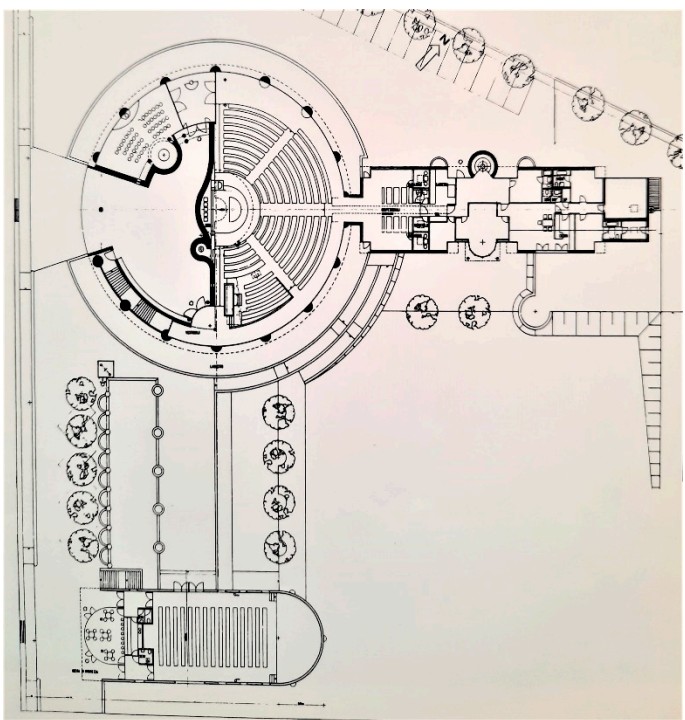

**Figure 2.** Plan of the complex. The pre-existing chapel is located at the bottom of the plan.

Justus Dahinden, who received a Catholic education, shared the thought expressed by Pope Paul VI in May 1964, when he convened a delegation of artists in the Sistine Chapel (Dahinden 1987, p. 15). At that time, the pontiff uttered his expectation to undertake a fruitful collaboration with them after centuries of any active role in the Catholic Church. Likewise, he suggested avoiding flashy choreographies, which was precisely why he invited the artists to use evangelization tools aimed and "rendere accessibile e comprensibile il mondo dello spirito, dell'invisibile, dell'ineffabile, di Dio. E in questa operazione [ . . . ] voi siete i maestri" (translation: "letting the world of the spirit, the invisible, the ineffable, of God accessible and understandable. By doing this [ . . . ] you are the masters") (Verdon 2008, p. 290). Gradually, this collaboration gave rise to responses both in the arrangement of the decorative elements and in the cooperation with the architects involved in the design of a type of building capable of responding to the contemporary cultural needs, especially in the archdiocese of Milan, whose cardinals proved to be open to the new architectonic tendencies (Santi 2020). During the developing process of the project, the patron asked the architect to consider the possibility of including the image of Christ in the main hall. As demanded, Dahinden produced a draft in which he envisaged a mosaic, fresco or bas-relief on the choir's wall representing Christ in an abstract form, but this was never realized due to financial concerns[5].

Justus Dahinden tried to convey the sense of community through the organic research that departed from the usual forms of traditional architecture. He proposed a particular way of approaching the divine, inspired by the latest trends and provisions in the design field. Truthfully, the post-conciliar strategies renovated the relationship between the priest and the assembly, which was supposed to feel like the protruding of God during the Eucharist. Everyone actively takes part in the mass (participatio actuosa), and the celebrant no longer turns his back on them as in the past, since the altarpiece is clearly visible in the

central point of the hall, not anymore leaning against the back of the nave in the presbytery (Benedetti 2000, pp. 107, 190). With just some gestures, the architectural conscience aimed at expressing the symbolic meaning as well operating in depth. In this way, the direct relationship between the assembly and the celebrant takes place (Santi 2020). Potentially, the space itself could give the necessary strength to the believers to aspire to holiness and fully grasp the message of an evangelical mission. The Belgian theologian Frédéric Debuyst depicted the transition to Modernity as follows:

> "Le mystère pascal pénètre ainsi la liturgie toute entière et l'anime jusque dans le détail de ses textes et des rites. Une caractéristique essentielle de la constitution sur la liturgie est précisément de rendre cette présence explicite et tangible à travers tous les types de célébrations: la messe, la prière officielle des heures, le sacrements". (Debuyst 1988, p. 8)

The architects were asked to propose spaces for the various devotional moments, and Justus Dahinden followed the recent parameters. This is the reason why from the initial stages, he divided the plan precisely according to the spoken worship (gathering function), with the Holy Communion as partaking (gathering function) and the individual meditation (devotional function) (Dahinden 1987, pp. 42–43). In his projects of chapels and Christian temples, these areas are always clearly outlined and produced after a profitable, rational process.

## 3. Entering the Transcendent

Notwithstanding that a real façade is missing in Saint Maximilian Kolbe church, its cupola awakes architectural associations with the mound or with the archetype of a shelter, creating a spiritual bond with the pre-Alps, too. Its shape exemplifies the celestial vault emerging from the water, a strong and pure symbolic cosmology (Petrini 1999). Faith is the only means through which we can perceive the transcendent, and the architect's purpose is to expedite this encounter, similar to an observatory fixing upon the stars.

Since the early drafts, which date back to spring of 1988 and are now part of the collection of the documents in the parish archive, Dahinden imagined a semi-sphere partitioned into different parts as a persuasive signal in the surroundings. However, the southern front hemisphere is wrecked; one half is occupied by the covered room, while the other one seems to be excavated in the volume and deprived of the roof (Benedetti 2000, p. 98). According to the Christian doctrine, God's perfection is evoked by the pure geometric form, but the building is deconstructed, as if the believers should only yearn to reach it: only in the new Life are we able to admire the sphere in its beautiful perfection. In this case, it is fragmented, and just two segments of the dome emerge to trace the access from the street, where the patio is positioned. The churchyard is an invariable element in the architectural tradition of the northern Italian churches, and Dahinden respected this building practice. It is the prelude of the revelation of the inner spaces, and that is why it is seldom separated from the street level by a few steps. It is a sacred area, but it is not included under the same roof.

From the churchyard in front of the building in Varese, two access doors lead to the vestibule and to the interior main hall. They are not visible from the avenue, and they stand at the corners of the court because the architect originally wanted the believers themselves to have to look for them. In my opinion, he emphasized the theme of the integration of the temple in the context, providing the city with an accessible space for free socializing which must be also recognizable in the distance, as he previously did in Monza with the piazzetta in front of the church entrance.

In Varese, the threshold separates the two worlds (the profane and the sacred ones), so it acts as a boundary distinguishing these dimensions. However, at the same time, it is the paradoxical place where these dimensions communicate. The only entrance to the church square is from the crevice in the white volume, which is emphasized by a solitary tree growing in the boulevard that naturally marks the gate (Figure 3). Doubtless, this is the most critical point of the project because it is the border between different realities, and

it has the task of attracting people inside the building and preparing them for the encounter with the divine. The temple has a different dimension from viale Aguggiari, so Dahinden imagined a sort of an open-air hall before the entrance itself.

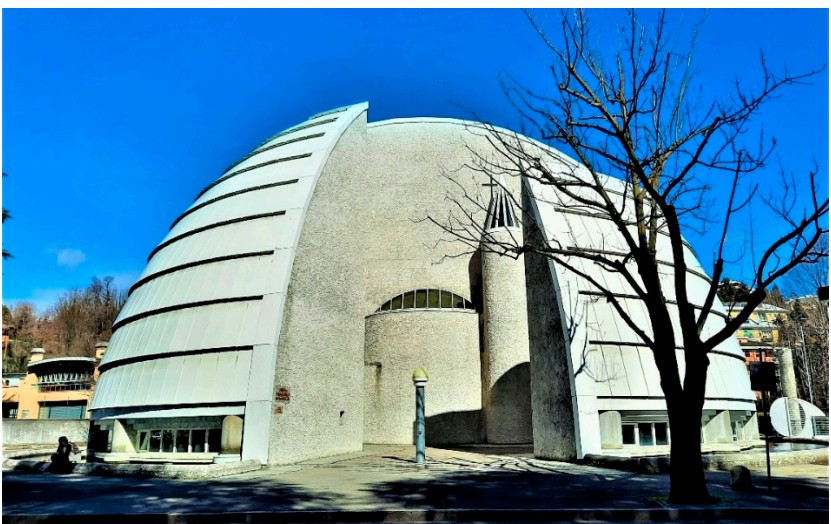

**Figure 3.** The southwest front of St. Maximilian Kolbe church, facing the avenue. Note the inner churchyard and the moat running around its base.

The main structure of the dome isolates the traffic noise and we have the impression of having stepped into a monastery cloister distant from the contemporary realm. In addition, the lack of bright colors encourages introspection and induces us to look up to the portion of the sky overhead. We immediately perceive the wind and the sound of the water, although it remains invisible from our point of view, as it is protected by the two wings of the building behind us. Therefore, the devotees can also meet the divine by experiencing in simplicity the natural elements, giving them the impression of being far from the city. During the warm season, the artificial pond which encircles the composition is not a mere decorative presence, on the contrary, it turns into a microcosm. There, water lilies, fish, turtles and occasionally herons coexist in harmony.

At this minute, Eden awakes a universal sense of life, and the dome itself represents the benign Nature rising up from water and, by some means, the architect depicted the genius loci of the Varese lake district, too.

In the churchyard, Dahinden implemented a potent and surprising gesture—almost contradictory compared with tradition—because he placed an 18-meter high blind wall in front of those who step into the churchyard that separates the outside from the inside of the edifice, as already described in the drawing section from 1 September 1988 (parish archive, Varese). The client did not immediately accept the proposal, but the architect was able to persuade the local community regarding the importance of the load-bearing wall, similar to a rough cliff rising out of water, providing stability to the entire composition[6]. At first, we feel a sense of alienation and discomfort that generates crucial moments of suspension, a sort of regenerating Christian catharsis. In the open-air room, we refine our senses, while we have the impression of being in front of a monumental Romanesque church with its tiny windows and strolled gray plastered wall (he also used plasterwork inside the building, as well as in other works, as happened in Le Corbusier's above-mentioned church). The irregular surface is put beside the brightness of the big shingled roof of the vault, which looks lighter and is positioned in a second moment to protect the vestiges of an ancient chapel. According to this clue, the three pure geometric volumes in the patio emerging from the walls look similar to the fragments of ancient buttresses, which had been included in the construction. We have the impression of standing in a metaphysical composition juxtaposed with the stereometric shape of the dome. The first volume is on our left, right below the three bells placed in the cavity left by the same cylinder, as if it has slipped under

the effect of the gravity, so that the entire yard acts as a sounding board, as suggested by the designer himself (Dahinden quoted in Servadio 1992, p. 73). Inside this section, the altar of the weekday chapel is enclosed, where the baptismal font is also placed next to the window, through which the moat can be seen. Water symbolizes the source of Life, as well as the purification of sin. The second volume is the largest one, and it emerges from the main wall that we have in front of us in the churchyard. This is the exterior of the bottom of the slightly pronounced nave behind the altar. Symbolically, zenithal lighting enters from the glass roof of the ledge, which remains invisible to the believers sitting on the benches inside the hall. The last one is the slenderest of the series and resembles a stylized bell tower with a circular plan, which actually contains the tabernacle and remains hidden from view, being the most sacred space in the Catholic liturgy, so only the minister can access it. Unconventionally, its position does not coincide with the post-conciliar liturgical dispositions, which wanted it to be visible by everyone. From the outside, the small recess remains protected by the glass roof, crowned by a small white cross, the only banner of Christianity visible from the outside. Even in the darkness, this symbol does not lose its meaning, since from the street it looks like a torch at night.

In the manner of a post-modern designer, Justus Dahinden played with the constitutive parts of the tradition. He deconstructed and reassembled them so the bells were not located in the belfry, containing the tabernacle, while the façade of the church was the back wall of the nave. The unconventionality in planning did not immediately persuade the client, who ignored the cross-pollination with the other international architectural episodes. Thanks to the minutes of the meeting on 27 February 1988 between Monsignor Arosio, don Brigatti, architect Dahinden and engineer Lorenzi, we became aware of these several difficulties. Preliminary doubts had been discussed concerning the overly modern tabernacle, as well as the visibility of the altar, which in the original scheme stood on the same level of the assembly and -in addition- the client expressed several hesitations about the blind wall in the patio, which would have been too "heavy"[7].

The examination of the preliminary drawings (1987–'88) reveals that Dahinden avoided any theatrical effect, as he choreographed a simple setting which would help the believers to take part in the liturgy. According to this vision, the tabernacle resembles a sepulcher, evoking Christ's Resurrection. Its unusual shape was the result of an inner debate between the architect and the client who asked to position it next to the altar. As a matter of fact, in the first stage, his proposal to put some mirrors inside the tabernacle to make it shine in the space was accepted[8]. However, during construction, the client changed his mind, since the daylight reflected on the bare, white surface of the round wall without any additional support (Servadio 1999, p. 512).

Actually, professor Dahinden was a fine connoisseur of history, so in the project, we can detect the memory of the severe pure forms of the Romanesque architecture, as well as the curved lines of the centrally planned Baroque chapels of the sanctuary of Sacro Monte having something in common with the graceful shape of the local lakes visible from the peak of Campo dei Fiori, the mountain located near the town[9].

The plan of the temple is entirely composed of the larger circular layout set against a rectangular smaller volume on the northeast side, where the rectory, the priest's apartment, the office connected to the parish archive, the rooms for the catechism classes and the confessional chapel are situated. The latter is accessible from the main hall by the worshippers, who come here to meditate before attending the main service. The human scale of the ambience results from the low ceiling and the semi-dark atmosphere, not too distant from medieval crypts, to improve introspection. The thin skylight running along the ceiling, through which the solar lighting enters, ends in the graft with the dome. In the respect of the original intent, access to the chapel opens from the hall and lies beyond the threshold, marked by two ancient bas-reliefs donated by the Milan cathedral (Figure 4). They echo the gothic spire placed on the concrete pedestal outside the building, a suggestion of the continuity between the ancient and the contemporary in the realm of Catholicism. Actually,

the two inner areas are not separated by any door to maintain the optical connection during mass.

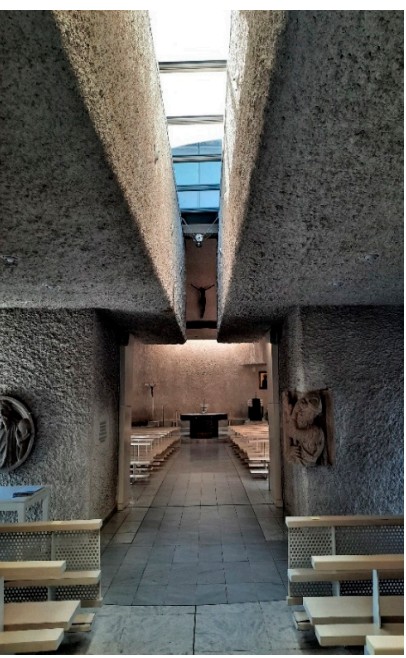

**Figure 4.** Perspective toward the altarpiece seen from the Chapel of the Confession. The vein of sunlight guides the devotees toward the main hall.

The devotion reserved for mobility is central in Dahinden's research, as it reflects liturgy, which is characterized by a series of gestures. Hence, the temple of Saint Maximilian Kolbe also must be discovered steadily, and thus the architect had taken into account the disposition of the inner paths. Once again, he respected the historical dimension and did not radically upset the spatiality. Whenever the celebrant reaches the altar from the vestry, he performs the introital procession before the celebration, and he remains partially lit up by the sunlight. This scenic detail denotes the theological value of the priest's role as a mediator, despite it being quite unique in the Italian panorama.

Furthermore, the architect's receptivity in exploiting the natural elements instills a metaphorical connotation and goes back to gothic architecture, when the primary importance was reserved for the lighting. After some time meditating in the chapel, the devotees are ready to follow the luminous vein that leads them without obstacles to the assembly gathered in the main hall, which greets them in an enlightened embrace of forgiveness. It is as if the light radiating from above enlivens the one that enters from the ribbon windows situated around the circular perimeter. What is more, after entering the church from the courtyard, we are lured into following an energetic circular itinerary that develops around the foot of the dome next to the glazed wall facing the moat, while the minister takes a direct route.

We can suppose that Dahinden took inspiration from the concentric orbits of the planets in the solar system, and it transformed into a sort of way to initiation before reaching our destination on an unusual spiral path, compared with the traditional ones along the church aisles. The circular shape of the plan and the slightly sloping floor toward the altar foster a sense of protection, which encourages participation in the ritual in progress (Figure 5). The inner disposition respects the tendency that had developed in Italy since the 1960s to abandon the usual elongated gallery, in which the celebrant and the believers were opposite each other, while it is in favor of them sitting around the presbytery (Benedetti 2000, p. 107). As a proof of this tendency, the Italian architect Glauco Gresleri observed that every person entering during the celebrations could be immediately recognized by the other worshippers (Gresleri 2021)[10]. In fact, the arrangement of the benches creates a sort of

an amphitheater with the altar as a focal point, and the singular member can be perceived as part of the community itself. Everything let us feel in harmony with ourselves in a genuine and communitarian dimension (ἐκκλησία). Even the light and the water that are integrated in the composition assume universal meanings, almost as if they are antidotes to the dull modernism, which often rules out the natural forms.

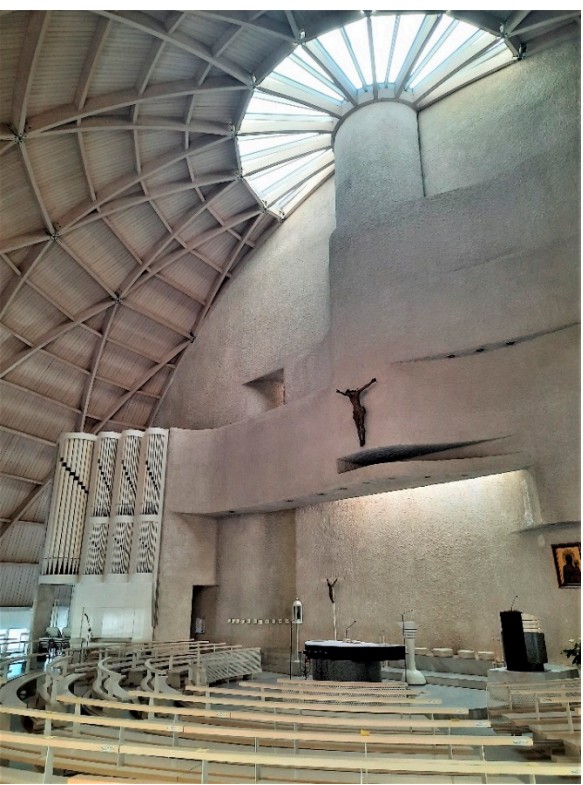

**Figure 5.** View of the semicircular main hall with the complex concrete wall. Above the altar, the oculus both physically and symbolically illuminates the assembly.

Consequently, the interior vault of the dome shows the prefabricated structural bleached woodwork, which is beautifully articulated, providing simple ornamentation and also giving a warm ambiance during the evening, when the lamps positioned at its base reflect the artificial light on its surface, as reported in the section of the dome in the detailed drawing from 12 February 1996 (parish archive, Varese), when the church was already in use[11]. It is interesting to observe that the Swiss architect introduced the use of wood, even though it was alien to the local building tradition, as was also noted by don Brigatti. However, this material belonged to Dahinden's personal background, like in Monza in 1973 (De Stefano 1983, p. 518).

The church is merely the house of God, so the light evidently has an allegorical and emotional implication. By means of this contrast, the architect is conscious that the divine dwells with the people attending the worship but not within the building itself. At its most basic, the temple hosts the repetition of Christ's Last Supper, and the congregation members are the apostles sitting around the table. Dahinden positioned it as the focal point of the space, despite the fact that it is not perfectly in the middle of the plan, the axis mundi in the Christian kosmos.

Justus Dahinden was particularly keen on researching the spiritual dimension in architecture, and in his mind, sculpture shared different points of contact with architecture. They both used the material as a medium for the transcendental dimension, and for this reason, the plastic nature of architecture emerges in most of his works. In the study case of the Northern Italian church, this trait had been accentuated and taken to the extreme

consequences. As a matter of fact, we have the impression of entering a plastic composition in the urban scale[12].

Actually, the drafts, the executive drawings and the photographs taken during the construction of the building witness his propensity to model the reinforced concrete as an authentic plastic composition, which is later covered by whitewashed plaster. The sinuous choir wall is the only load-bearing element in the construction. It is literally the "cornerstone" on which the entire temple stands, so it has a metaphorical meaning and exudes a rare expressive quality. It does not only separate the outside from the inside, but the protuberance behind the altarpiece incorporates and protects the tabernacle, and the celebrants' seats are integrated into the perforated surface, too. During the day, the solar light passes through the glass roof above the bay, while light sources are hidden from the worshippers' point of view, with the aim of giving prominence to the Holy Mystery rising from the sky. Inside the space, this wall contributes to producing an internal landscape, as it evokes three-dimensionality on its own. It almost appears to "come to life", with a kind of "wave" defining the choir area and constituting the tribune leaning against the white organ (manufactured by the Mascioni company in 1997), which became relevant after the Second Vatican Council, as musical accompaniment became an important instrument for the celebrations (Benedetti 2000; Verdon 2008, p. 350). By resorting to this solution, the upper part of the wall ends with a cylinder-like volume marking the highest point of the structure, like a stylized spire or a telescope. It resembles the provision of a wide fixed curtain covering the community, from which a series of "tie rods" supporting the structure unravels and forms the woodwork ribs. Precisely at the point where they hinge, a crescent skylight (or oculus for the telescope) opens, through which zenithal light slides along the rough surface of the underlying wall, giving an intense atmosphere to the composition (Figure 6).

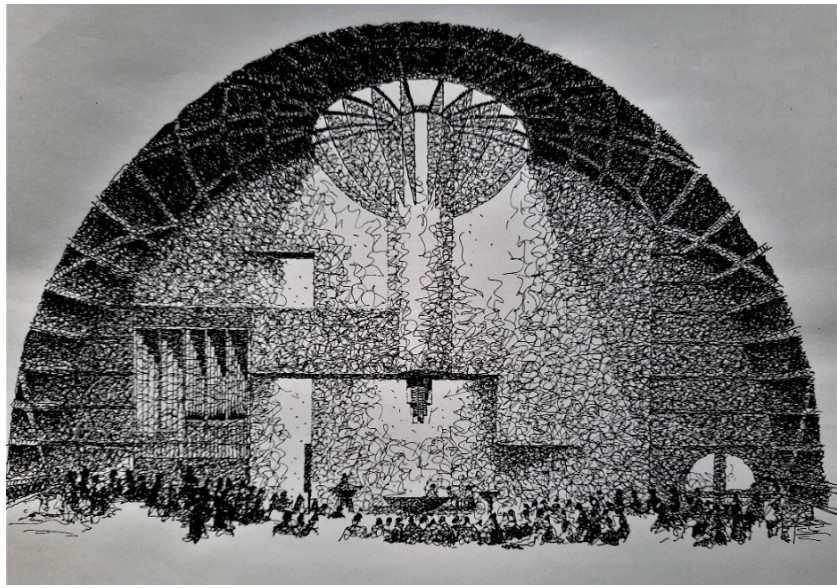

**Figure 6.** Justus Dahinden's undated pen sketch representing the symbolic light entering through the skylight positioned above the altarpiece (parish archive, Varese).

By entering the semi-dark lobby and passing through the white door, we reach the main space, which triggers emotions. Daylight pours in through the ribbon window situated along the base of the dome, and hence we have the impression it derives from the lowest part, as if a gust of wind had moved the "tent". It separates the covering structure from the soil to give the chance to discern the outside moat. The church is lit by the long ribbon window, and as a result, the natural light reflected on the clear floor encourages people to linger. Thanks to this solution, the structure radiates a sense of dematerialization, as it seems to be neither anchored on the ground nor positioned on a drum. It is acceptable

that the cupola appears to "fluctuate" because the religious belief and the faith let the believers consider everything beyond the scientific physical demonstrations.

The passage between the outside world and the inner one is nuanced through the extent of the luminosity. However, the glazed walls measure less than a meter high to obstruct the view on the landscape and spread an indirect radiant light on the floor. Dahinden stated that churches had to be introverted spaces for intimacy, so they must be sober in decoration and preclude any vis-à-vis with the surroundings (Dahinden 2005, p. 74). Meanwhile, the light guides us along the perimeter of the building before finding space at one of the 500 seats on the benches. By walking along the hall, we understand that its plan is the empirical result of accurate research, in which we are the main characters with respect to the idea of active participation. The architect tried unconventional forms with the purpose of fulfilling the practical and spiritual requirements.

### 4. The Cupola as a Reflection of the Universe

What did the shape of the dome mean for Dahinden? Why did he choose this form for the plan of a Catholic church? By referring to his previous works, he estimated it to be the most adaptable solution for some typologies of spaces, whose function, in this case, is to gather the worshippers under the same roof. Although this gesture seems to be radical (but not provocative), the architect paid tribute to the masters' tradition, since the cupola is the universal emblem of the historical temples. Nevertheless, it is more here: "the space, the sky, the earth are spherical"[13], declared Dahinden significantly. Since the early drafts of the church produced in spring of 1988, now in the parish archive in Varese, he had in mind the kosmos and the celestial vault as a deep concentration of significances. One of the repeated qualities of his design interventions—especially in those of a religious nature—is the search for the spiritual dimension accessible to all. Once, the Danish architect Steen Eiler Rasmussen wrote that "art should not be explained; it must be experienced" (Rasmussen 1964, p. 9), thus, it seems that the Swiss colleague relocated the same concept into architecture. In his projects, Dahinden looked for the gestalt that he eventually found through various formal solutions of a spatial type but also a practical one, such as the familiarity with the modern materials and techniques. As a matter of fact, in the case of the Italian church, both the client and the designer aspired to give birth to a temple which would have been accessible forever (Brigatti 1997a, p. 108).

The space of architecture belongs to the city, so churches are no longer built as monuments but rather as places to be experienced tout court. There, the believers should take on the environmental conditions, accentuate them moderately and deepen them spiritually.

As was already mentioned, the main reason for the use of the dome in the church is that the hemisphere encourages connotations with the sky, and it is universally accepted as the emblem of the perfection of God. This leads us to another consideration. Despite the fact that the client was not immediately convinced to whitewash most of the surface of the building, Dahinden was able to persuade the commission that the white color represents a form of suspension, of divine immanence that must itself be detached from the context[14]. The sunlight reflected on the rough surface of the walls, on the smooth floor and above the vault gives birth to shimmers, which make the space intriguing, as if Nature is entering the building. From this point of view, it is clear that the architect did not consider the color to be of minor importance. This is the reason why, in his latest ecclesiastical projects, he entirely painted with clear colors the inner ambiences of the buildings, although in Varese, Dahinden did more by also using white (as the sum of all colors) outside the dome to highlight a place of salvation, distinguished with its massiveness. White can also be seen as a symbol for the new millennium, perfect for the search for another dimension beyond the time and the surrounding space but also an impeccable folio destined to be used as a support to write the "new story" of the reborn community[15].

In the bargain, Justus Dahinden's predilection for the mega-structures in which obliquity prevails can be a direct effect of his aversion for verticality (Dahinden 2005, pp. 26–33). In this project, his attempt to circumvent the monumental effect at all is evident, but on

the other hand, he wished to generate an intimate atmosphere. In his mega-structures, he rejected any colossal vertical outcome that was perceived to be "inhuman" (Dahinden 2005, p. 53). On the contrary, he was in favor of the diagonal in space, judged as a much more welcoming form. The result reflects the cultural contamination with the mid-century avant-gardes, such as the British Archigram and the Metabolist group in Japan, who stressed the accent on the idea of organic growth (Dahinden 1987). Still, a few years later, Dahinden came with his mind on those experiences while he was developing the plan of the Italian church. During his long career, the professor experimented with the dome as an archetypical form in different contests. We found it as the model for one of his utopian urban plans, too. In this case, through the scheme, he proposed an integrated urban and residential system in the form of a hill, on which an entire town with leisure activities as well as private housing were supposed to be built. The Urban Mounds projects pursued a better quality of life for the inhabitants in minimized distances from the different sites, which were located in a multifunctional structure associated with a green mountain. The expressed sense of community remained a constant in his works[16].

In agreement with his personal conception, architecture consists of two points—the physical and psychological parts—which influence the behavior of individuals (Dahinden 2005, p. 105). Reaching the synthesis between them was inspiring, so he considered the emotional and psychological spheres essential in the definition of the space. One of the greatest criticisms directed toward contemporary architecture is its lack of gestalt, because often the buildings are not able to stir feelings. Dahinden's main aspiration was to produce something that could be perceived as an expressive and unique element in the urban contest, aimed at attracting citizens. Likewise, this is the main reason why he included the cupola in the composition (Servadio 1999, p. 510).

In contrast with the fact that much of the structure was prefabricated and assembled on site under the supervision of the local engineer Ezio Lorenzi, who had been involved from the beginning of the project, the care for details is evident. Similar to a skilled craftsman, Justus Dahinden paid attention even to the most minute element and checked the progress of the work while he gave advice to the workers he met during his frequent inspections. The colleagues and the parishioners who assisted him described his profound empathy and efficiency in collaborating only by using his pencil and sketchbook "like a patient father accompanying his growing son"[17]. His extinguished cigar in his mouth and the benevolent smile peeking out under a moustache expressed his satisfaction for the job. That is the reason why, at the end of his assignment, he declared "of all my works, my favourite is the church of Varese" (Dahinden quoted in Servadio 1992, p. 73).

## 5. Conclusions

In light of these reflections, we can consider that the project of the parish church of Saint Maximilian Kolbe in Varese represents the conclusion of a long period of research which received a hint from the post-conciliar liturgical reform. Nevertheless, thanks to the profitable dialogue which Justus Dahinden had with the enlightened clientele (underlined by the correspondence between the two parts), he realized one of the most off-the-beaten-path examples of sacred architecture at the end of the millennium. At the same time, he kept in mind the demands and necessities of the future devotees at the forefront.

In order to lead the worshippers closer to the manifestation of the transcendent, the designer opted for the inclusion of natural elements, primarily water and light (the latter is not considered an immaterial element, vibrating on the surfaces), as a sort of pantheistic revelation of God. For example, daylight enters from different directions, being warm and direct from above while indirect light is mediated by the surface of the water embracing the feet of the dome outside.

Since the early drafts, we have become aware of the fact that the architect refused any theatrical effect. On the contrary, he opted for architectonic laconicism by using a few elements in simple spaces, realized by combining elementary forms and materials. The renewed Catholic liturgy played an important role in the development of the plan,

since both the client and the designer reflected on the function of the building itself. In its essential form, liturgy is a series of symbolic gestures which had been considered during the genesis of the master plan. As a matter of fact, the architect carefully analyzed this aspect, and he differentiated the path followed by the celebrant from the devotees' one. Sacred architecture is the product of long cogitations, and in this case, it is the response to special liturgical needs. The space itself became the epiphany of the Christian community, being joined together in the cityscape.

More care had been given for the emotional aspects, too, which are fundamental to sealing up the encounter with the divine. As a talented director, Dahinden managed to generate unexpected instants of suspension before revealing the encounter with the divine in order to lead the believers to the optimal condition, like during a pilgrimage. Therefore, the use of natural elements contributes to this purpose. Here, empty spaces play a seminal role both outside and inside the Italian complex. The churchyard seems to be carved out like a valley in the rock, while the main hall stands in the core of the "hill", as if the architect wanted to identify God as the axis mundi, of which the altar epitomizes the visual center. Perhaps the most difficult obstacle the architect from Zurich had to face was the request to draw a setting in which both material and non-material aspects, as well as rational and irrational elements, were included without any banal solution.

Furthermore, a good church project is not defining a perfect conclusion, but the one that comprises the dynamism of the community life and complexity of the relationships of its members and which can absorb spontaneous changes, too. In Saint Maximilian Kolbe parish church, the worshippers are invited to conduct a profitable dialogue with the celebrant, because they are not separated by any barrier as in the past (the presbytery is missing). All in all, here, everyone can experience the joy of listening to the Word of God together with others. As a consequence, this bonds the idea of being part of the same community in the "veritas rituum" (Benedetti 2000, p. 61). As soon as the architect was commissioned to draw the church and the parish center in Varese, he had the chance to enhance everyone's lives far from the daily rush, but as a place destined to give release and to make possible the encounter with the spiritual. He shared with the client the desire to "ritorno della Chiesa all'essenzialità dei suoi elementi costitutivi, cioè al suo essere una comunità intorno alla Parola e all'Eucaristia"[18] (Apa et al. 2004, p. 280).

In the project of the Saint Maximilian Kolbe church, Justus Dahinden demonstrated that the transcendent might be included within contemporary architecture. The designer refused any theatrical or monumental solution, as the key was the simplicity of the materials (wood and reinforced concrete), the new spatiality derived from the models of the past (central plan) and the elementary nature of the forms (cupola). All those reflect the renewed liturgy and revitalized the relationship with the transcendent. Architecture succeeds in expressing the supremacy of the transcendent without forgetting the devotees' needs by recollecting them as well. Here stands Justus Dahinden's main contribution: giving spiritual qualities to a sacred, contemporary building.

**Funding:** This research received no external funding.

**Institutional Review Board Statement:** Not applicable.

**Data Availability Statement:** Not applicable.

**Conflicts of Interest:** The author declares no conflict of interest.

## Notes

[1]　Justus Dahinden (1925–2020) graduated from the Polytechnic of Zurich in 1949, studying Architecture. In 1955, he opened his office in Zurich, while in 1974, he became a professor at the Polytechnic in Vienna, where he taught interior design for 21 years. Since the 1960s, he was considered one of the most active architects in his country, with a long list of works and publications. As it happened for the majority of the architects of the time, with no exception for him, Le Corbusier was estimated to be an unattainable model, but his early projects show his personal touch. In fact, Justus Dahinden highly considered the emotional part of his plans. In his opinion, a building had to be perceived not only as a physical entity but as something beyond its own appearance. For this reason, he inaugurated research by using abstract and pure geometric volumes in his plans

in Switzerland and abroad. He mainly focused on urban proposals, touristic buildings, housing and churches and utopian experiments, too. During his long career, in 1956 he published his PhD dissertation entitled *Versuch einer Standortbestimmung der Gegenwartsarchitektur* (Attempt to Determine the Position of Contemporary Architecture) and various essays in which he described his modus operandi and expressed his personal opinion about his mission as a designer. He died in April 2020 at the age of 94. His drawings, scripts and publications had been donated to the Institute für Geschichte und Theorie der Arkitektur (ETH) in Zurich and are collected in the FRAC Centre-Val de Loire, Orléans. They are accessible to scholars.

[2] Letter dated back to 17 September 1987 (parish archive, Varese).

[3] Translated by the author: "(a) would you prefer an exclusively reserved for the religious service building, though traditional (b) in harmony with the landscape, despite more expensive?"

[4] The analysis of the correspondence and the documents collected in the parish archive in Varese revealed that on a primary stage, the local commission considered giving the assignment to two other candidates: the Italian architects Carlo De Carli and Antonello Vincenti, who both designed churches in Milan in the 1950s, although in a second moment, they opted for Dahinden, whose work had been highly appreciated for its originality.

[5] The sketch was included in the letter sent to don Brigatti on 10 October 1989. Due to financial restrictions, the images had not been realized (parish archive, Varese).

[6] Letter sent from don Giovanni Brigatti to Justus Dahinden in 1988 (parish archive, Varese).

[7] The minutes of the reunion have been collected in the parish archive in Varese.

[8] Letters sent from don Brigatti to Dahinden on 6 October 1992.

[9] His sensibility toward the environment is acknowledged, as he demonstrated through his fulfilled projects and by lecturing around the world. For instance, both the Rigi Haus (Rigi) and St. Stephan church (Arlen) are protected by pitched roofs in the respect of the alpine constructive techniques, while in Iran in 1979, he developed an earthquake-proof design based on the local dome structure. On the other hand, in east Africa, in 1973 he drew the plan of the new Catholic church of Mityana. There, he did not imagine any belfry, which would have been alien to the local inhabitants, but he included a tower from which they could play can drums (Dahinden 2005, p. 39). Furthermore, as a talented craftsman, he relied on his sensibility for the use of regional materials.

[10] This respected the original intention, despite don Brigatti not immediately being persuaded by this proposal, as he expressed during the meeting in February 1988 (see Note 7).

[11] The architect also drew the lamp standing in the churchyard. The sketch dates back to 29 March 1994 (parish archive, Varese).

[12] From the beginning of his career in the late 1950s, Dahinden subverted the regulations of the existing building codes. He wished to probe unexplored fields that potentially could offer stimuli to the development of tomorrow's architecture, as he highlighted in the majority of his scripts. He cultivated the interest for the mid-century utopian avant-garde and used sculpture to cross the boundaries of the architectural rigor and the barren functionalism. In addition, he improved his skills in manipulating both traditional and synthetic materials, in the manner of a talented artist. In the 1970s, he carried out some experiments. He created unconventional pneumatic structures, a capsule-like bubble house system in the desert areas in the Middle East and designed the floating theatre, similar to a flying saucer, gliding over the surface of the water of the lake of Zurich (Dahinden 2005). He remained faithful to that research and his "Philosphie der Schräge", the "philosophy of inclination" (https://www.tagesanzeiger.ch/der-pyramidenbauer-ist-nun-im-himmel-166690949717, consulted on 21 November 2021), and took the plastic research—which culminated in the church of Saint Maximilian Kolbe—to its most radical significance. These experiences enabled him to define unexpected silhouettes, of which we can find hints in the theme of the Italian megastructure.

[13] Dahinden quoted in Servadio (1992, p. 73).

[14] In an undated note, don Brigatti suggested the use of terracotta tiles or slates. They are both materials which belong to the local building tradition (parish archive, Varese).

[15] Widespread difficulty in accepting ascetic architecture that has no canonical form, but elements alien to the local culture cosmic form, for which the extensive use of white has incurred the general disapproval of the local community. The members did not immediately appreciate this "emptiness", but with the passing of time, they became more familiar with it.

[16] Another stimulating unrealized project in which we find a dome is the Floating Theatre, which had been commissioned by the Werkbühne theatre in Zurich in 1970 (the drawings are digitalized and accessible on the website of the FRAC Institute, Orléans). The architect created a structure with a bubble-like shape, as if it were an alien flying saucer gliding over the surface of the lake of Zurich. The structure was supposed to host about 500 spectators, and a circular foyer could be turned into a temporary art gallery, while the upper glass dome could be opened if needed for the shows. Once again, the professor treasured the natural element, which was the real protagonist of the project.

[17] Personal communication with Mr. Guido Negrini, 25 May 2021.

[18] "the return of the Church to the essentiality of its basic components, to its being a community gathered around the Word and the Eucharist" (translated by the author).

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
