# Peer review of "Cosmology, Faith, Architecture—A Temple under the Sky: The Church of Saint Maximilian Kolbe in Varese"

_religions, doi:10.3390/rel13020111_

Round 1

Reviewer 1 Report

It is applaudable that the author brings this nice piece of modern church architecture to the attention of a wider international audience. However, I think this article lacks a real and clear research question, which also prevents drawing solid, academically interesting conclusions. It is a very descriptive piece, with very few critical or analytical reflections. It is also clear the author admires the building and the architect, which is not a problem as such, but it becomes a problem when there is no single line of critical engagement with the subject. I also found several spelling mistakes, and particularly the too abundant use of capital letters, the wrong use of the word 'contest' (instead of, I think, context), and the also capitalized but non-inclusive use of the word 'Man' for humanity, are disturbing. I also think the references made to Vatican II and the liturgy are too shallow and should be deepened more, but this also depends on the research question. Mention is made of interviews with parishioners, which gives the impression of some kind of empirical research, but there is no access or reference to clear data or explanation of the research method.
The author convinced me this church building and the intentions and motivations of the architect are really interesting to report, but they need to be structured as an answer to a clear research question, have a good methodological underpinning, and be presented as clear argumentations leading to a specific conclusion. 

Reviewer 2 Report

  1. more detail on community involvement and how it has affected the definition of the project (lines 79, 88);
  2. to better understand some aspects (e.g.,lines 146-149; 165-170; 212), it would be useful to insert a plan of the entire project;
  3. the text would benefit from greater narrowness in some passages of the hermeneutic approach (e.g.,lines 274-276; 280-281; 307-308);
  4. it would be interesting to add information on the letters between the architect and the Italian priest; if possible, clarify how this report modified the project (line 233);
  5. In order to clarify the genesis of the project, it would be useful to deepen (even with bibliography) the development on the inner disposition of the celebrant (lines 284-287);

Round 2

Reviewer 1 Report

I see you tried to take the feedback into account, but I think the research question can still be more clear and should be more leading throughout the article. The English also needs a last check. I'll leave it to the editors to make a final decision, but I do think the article has improved a lot in comparison to the first version.

Author Response

Dear Reviewer,

thank you again for your kind feedback. I will be waiting for the editors' final judgment.

Regards,

Author